# Structural Disparity of Avocado Rootstocks In Vitro for Rooting and Acclimation Success

**Jayeni Hiti-Bandaralage ***, **Alice Hayward** and **Neena Mitter**

Centre for Horticultural Science, Queensland Alliance for Agriculture and Food Innovation, The University of Queensland, Brisbane, QLD 4068, Australia
* Correspondence: uqjhitib@uq.edu.au; Tel.: +61-469-869-017

**Abstract:** Improving clonal rootstock propagation of avocado has been a major industry/research challenge globally for many decades. Tissue culture has been a focus for clonal propagation with substantial advancements in recent years. In the process of tissue culture, avocado rootstocks display differences in rooting and acclimation capacity. Such differences may relate to the specific structural characteristics of the rootstock. This study aimed to investigate the structural difference during tissue culture in two rootstocks 'Reed' and 'Velvick', with differing rooting and acclimation capacity. Histological investigations were carried out of stem vasculature, leaves and roots of tissue cultured plantlets. Quantitative parameters; stomatal index, stomatal density, trichome density, vein-islet density and vein termination density were also analysed. Prominent fascicular cambium and fewer phloem fibres in stems positively correlated with rooting capacity. Acclimation success positively correlated to the presence of fully differentiated secondary xylem in root. Presence of smaller epidermal cells, high stomatal density and reduced vein termination density was associated with reduced acclimation success. These findings will support optimisation strategies for micropropagation of not only difficult-to-root and difficult-to-acclimate avocado rootstocks, but also, other woody perennials experiencing similar problems.

**Keywords:** histology; avocado; 'Reed'; 'Velvick'; micropropagation; in vitro; adventitious rooting; acclimation; difficult-to-root; difficult-to-acclimate

## 1. Introduction

Vegetative propagation is the sole approach to preserve genetic homogeneity in open pollinated plants during propagation. This requires adventitious root induction on cuttings or on in vitro cultured/tissue cultured shoots by exogenous application of rooting hormones, mainly auxin [1]. In either case, adventitious root induction is difficult and rate-limiting to the propagation of many woody fruit species [2,3].

Avocado (*Persea americana* Mill), belongs to family Lauraceae, is one such difficult-to-root perennial consistently demonstrating poor adventitious root induction, especially after attaining the mature stage [4–8]. In the process of clonal rootstock propagation, 'budwood' (cuttings) are grafted to a nurse seedling and subjected to uninterrupted dark period (etiolation) to produce elongated white shoots [9]. At the end of the etiolation period the base of the elongated shoots is scrapped, and rooting hormone is applied to induce rooting. This current nursery practice is resource intense and a cumbersome operation at commercial scale [10].

The rooting stage is also the most critical and rate limiting step of avocado tissue culture/in vitro micropropagation [5,11–14]. This has been reinforced over many years of our research in developing a high throughput micropropagation system for avocado using physiologically mature explants [7]. Most importantly, the rooting capacity of in vitro regenerated shoots was highly influenced by genotype. Our research identified that the avocado rootstock 'Velvick' is extremely difficult-to-root compared to rootstock 'Reed'.

Additionally, the downstream acclimation process is also very challenging for 'Velvick' with only 40% survival compared to 'Reed', in which 97% survival rate could be achieved [15].

The anatomical plasticity at a particular growth stage can affect adventitious rooting in two ways; either by affecting formation of root initials or by affecting root emergence. Structurally, the latter often correlates to increased secondary growth, commonly found in stems for mechanical support [16]. Favourable structural features (active vascular cambium, less sclerenchymatous phloem fibre [16] and adequate energy (sugars or starch) resources are correlated to improved adventitious root induction [17]. Therefore, easy-to-root and difficult-to-root genotypes may display distinct structural differences. When plants are grown under in vitro conditions it is stress free, hence can alter the inherent morphological features in plantlets, including essential structural characters vital for survival under dynamic natural environments [18]. In particular, the leaf and root structures developed under in vitro conditions largely affect the success of acclimation process [18]. The hardening or acclimation process is the stage where in vitro plantlets are gradually exposed to normal environmental challenges, thus, are forced to develop complete survival mechanisms for a dynamic septic environment.

The current study focused on a histological investigation to provide possible explanation for differences in rooting and acclimation capacity of two avocado rootstocks subjected to tissue culture. Structural attributes of in vitro shoots and acclimatising plants of each rootstock were compared. Stem sections from glasshouse grown plants (non-tissue cultured) were also included to compare the general stem morphology of two genotypes. Further, the structure of etiolated shoots of 'Velvick', used for root induction in the commercial clonal propagation was studied to relate structure for improved rooting capacity.

## 2. Materials and Methods

The anatomical structure of stems, leaves and roots were compared for avocado rootstocks 'Reed' (easy-to-root and acclimatise) and 'Velvick' (difficult-to-root and acclimatise) under in vitro and ex vitro conditions as follows:

### 2.1. Stem Micromorphology

To associate the morphology of avocado stems with capacity for adventitious root induction, transverse stem sections of rootstocks 'Reed' and 'Velvick' were taken from in vitro elongated shoots prior to rooting. Sections were also taken from actively growing softwood stems of glasshouse-maintained physiologically mature plants (5 years from grafting) during late summer (February) for comparison. Also, for 'Reed', in vitro shoots after 4 weeks from rooting treatment were sectioned to observe root emergence.

In vitro shoots for both rootstocks were obtained from the proprietary tissue culture system developed by Hiti-Bandaralage et al. [7,10] at University of Queensland, Australia. Cultures were from mature shoot explants initiated at the same time and maintained for 10 subcultures (4 week intervals) in order to provide identical shoots for analysis. Glasshouse-maintained plants comprised mature budwood from 'Reed' and 'Velvick' rootstocks grafted onto seedling rootstocks and maintained for more than 6 years in The University of Queensland glasshouses under natural daylength conditions at an average of 27 °C/24 °C day/night, 70–80% humidity, 12 h day length.

Mature avocado stem (budwood) material is conventionally rooted during clonal avocado nursery propagation using the Frolich and Platt propagation method [9,19]. During this process, two-weeks of dark incubation, to etiolate growing shoots destined for rooting, is an absolute requirement for rooting success. Root induction in difficult-to-root rootstocks is impossible without this etiolation step. The transverse stem sections of etiolated and de-etiolated shoots were thus hypothesised to provide hints as to the anatomical basis for the high rooting capacity of etiolated shoots vs. de-etiolated shoots. To determine the effect of etiolation on stem morphology, stem sections were taken from etiolated and de-etiolated 'Velvick' plants produced using the traditional commercial nursery propagation practice (Anderson Horticulture, Duranbha, NSW, Australia). This involved grafting

mature 'Velvick' budwood onto a nurse seedling, followed by a two-week dark incubation to etiolate shoots prior to sectioning. A subset of etiolated plants was then de-etiolated under a 16 h photoperiod in a laboratory growth cabinet for eight weeks, prior to sectioning.

Stem Sample Preparation and Observation

In all cases, a 1 cm portion of stem tissue was sampled 2.5 cm below the shoot tip from five plants ($n = 5$). The pieces were embedded in 6% agar and mounted on a vibratome (LEICAVT 1200S) to cut 50 μm sections. Sections were placed on a glass slide and treated with a drop of chloral hydrate solution (saturated in lactic acid). Slides were incubated for 30 min at 50 °C in a water bath enclosed in a Petri dish. Sections were then washed with distilled water, stained with 0.5% (*v:v*) Safranin O (dissolved in 50% ethanol) for 3 min followed by de-staining; first with 70% ethanol for 1 min (3 times), then with 100% ethanol for 1 min. Finally, the sections were rehydrated, mounted with water and observed under compound photomicroscope (Nikon Y-FL, Manufactured by Nikon Instruments Inc., USA supplied by Coherent Scientific, Adelaide, South Australia). Photographs were taken using a camera head (Nikon DS-Ri1, Manufactured by Nikon Instruments Inc., USA supplied by Coherent Scientific, Adelaide, South Australia) fixed to the microscope with the help of NIS elements basic research software.

### *2.2. Foliar Morphological Study of Avocado*

To associate the micromorphology of the leaf epidermis with acclimation success in 'Reed' and 'Velvick', sections were taken from in vitro rooted shoots at de-flasking and at two acclimation time-points: (A) 5 days after de-flasking, prior to potting out for acclimation, and (B) 1 month in acclimation. Number of stomata per field, number of epidermal cells per field, and number of trichomes per field were counted on the abaxial side of the leaves (one field per leaf, two leaves per plant to get average value per plant) to obtain average values for 10 plantlets ($n = 10$) at de-flasking and 1 month in acclimation. Differences were assessed by calculating the stomatal index, stomatal density and trichome density. Stomatal index was calculated by: Average number of stomata per field/(average number of stomata + average number of epidermal calls) × 100. Stomatal density and trichome density were a measure of number of stomates or trichomes per 1 mm$^2$ of leaf area.

Leaf architecture was also compared with respect to vein-islets and vein termination for the rootstock at in vitro and acclimation stages. Number of vein-islets and number of vein terminations per 1 mm$^2$ leaf area were calculated for 10 samples in averaging values from two leaves per plant.

Leaf Sample Preparation and Observation

The second and the third fully expanded leaves were sampled from ten plantlets for in vitro (rooted), de-flasked and acclimated stages. For epidermal observations, clear nail polish was applied on the abaxial surface of the leaf and let to air dry. A clear sticky tape was then used to peel off the dried nail polish layer adhering to lower epidermis. The preparation was mounted on a slide, observed under a compound light microscope (Nikon Y-FL) with ×400 magnification (objective ×40, eye piece ×10) and photographed using Nikon DS-Ri1 camera.

To study the vein architecture a leaf fixation protocol by Mani et al was followed (incubation times were modified) [20]. Excised whole leaves were fixed in a 1:1:3 (*v:v:v*) solution of formalin, acetic acid and 70% ethanol for 24 h. The fixed leaves were then incubated for 24 h in 70% (*v:v*) ethanol to remove all chlorophyll followed by bleaching in 5% aqueous solution of NaOH for 3 days. Prepared leaf samples were washed several times with water, blot dried and immersed in saturated solution of Chloral hydrate in lactic acid for another 3 days. Cleared leaf tissue were then washed with distilled water and stained with 0.5% (*v:v*) Safranin O (in 50% ethanol) for 10 min followed by de-staining, first with 70% ethanol for 10 min (2 times), then with 100% ethanol for 30 min. Sections of 2–3 mm$^2$ were cut, rehydrated and mounted on a glass slide with water. The preparation

was observed under compound light microscope (Nikon Y-FL, Adelaide, South Australia) under ×40 magnification (objective ×4, eye piece ×10) and photographed using Nikon DS-Ri1 camera.

### 2.3. Root Structure of In Vitro Cultured Avocado

Transverse sections of the roots of in vitro rooted shoots of rootstocks 'Reed' and 'Velvick' were observed for comparison. Sampling was done 5 days from de-flasking, prior to potting for acclimation.

#### Root Sample Preparation and Observation

A 0.5 cm section of root, ~1 cm upstream of the root tip, was sampled from 5 rooted plantlets for each rootstock. The pieces were embedded in 6% agar and mounted on a vibratome stage (LEICAVT 1200S) to cut 50 μm sections. Sections were placed in a Petri dish and incubated with few drops of chloral hydrate solution (saturated in lactic acid) at 25 °C for 30 min. Sections were then washed with distilled water, stained with 0.5% (*v:v*) Safranin O (dissolved in 50% ethanol) for 3 min followed by de-staining first with 70% ethanol for 1 min (3 times), then with 100% ethanol for 1 min. Finally, the sections were rehydrated, mounted with water and observed under photomicroscope (Nikon Y-FL, Adelaide, South Australia).

### 2.4. Data Analysis

Quantitative data, stomatal index, stomatal density, trichome density, vein-islet density and vein termination density were analysed using the statistical software IBM SPSS 23. Data were normally distributed according to the Shapiro-Wilk test and Q-Q plots. Homogeneity of variance was in accordance with Levene test. Analysis of variance was conducted, and multiple comparisons were done using Tukey HSD to identify significant differences between groups.

## 3. Results

### 3.1. Micromorphology of Avocado Stem

#### 3.1.1. Morphological Alterations during Tissue Culture

A remarkable difference in stem morphology, at the region routinely treated to induce adventitious rooting (2.5 cm from the shoot tip), was observed between in vitro 'Reed' and 'Velvick' shoots, which show marked differences in rooting competency (Figure 1). 'Reed', the rootstock with a high rooting competency (100% rooting with the optimum treatment) under in vitro conditions revealed a stem structure favourable for adventitious root induction. In vitro 'Reed' shoots possessed a prominent continuous fascicular cambium of 3–4 cell layers (Figure 1a). Xylem rays were directly opening onto the cortex and a negligible amount of sclerenchymatous phloem fibres were present. In addition, the primary xylem tissues were well developed with large vessel element layers (Figure 1a). In contrast, in vitro 'Velvick', which is difficult-to-root (33% rooting with the optimum treatment) did not display a continuous whirl of fascicular cambium but had a large amount of phloem fibre with secondary thickening layers (Figure 1b). Though not related to rooting capacity, lesser amount of large vessel elements was observed in primary xylem layers of 'Velvick' stem section (Figure 1b).

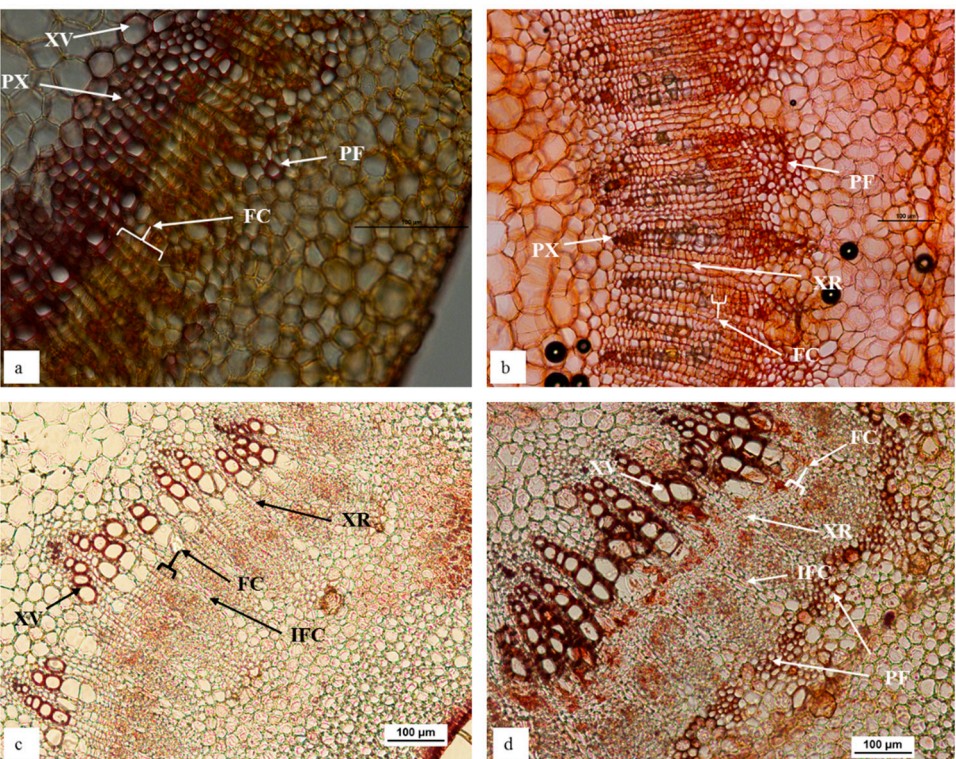

**Figure 1.** In vitro 'Reed' stem is comprised of prominent continuous fascicular cambium, negligible amount of sclerenchymatous phloem fibres and large vessel elements in primary xylem tissues (**a**), in contrast to in vitro 'Velvick', which displayed a staggered fascicular cambium, highly abundant with phloem fibre and lesser amount of large vessel elements in primary xylem (**b**). Softwood stem of 'Reed' had wider continuous fascicular cambium with no phloem fibre (**c**), while 'Velvick' possessed comparatively narrow continuous fascicular cambium and a nearly complete ring of phloem fibre (**d**). Sections obtained from avocado stem at 2.5 cm below the shoot tip: in vitro stem of 'Reed' (**a**), in vitro stem of 'Velvick' (**b**), softwood stem of glasshouse mature plant 'Reed' (**c**), softwood stem of glasshouse mature plant 'Velvick'. Stems were embedded in agar, sectioned at 50 μm, cleared with chloral hydrate and stained with Safranin O (*n* = 5). PX = primary xylem, XR = xylem ray, FC = fascicular cambium, IFC = inter-fascicular cambium, PF = phloem fibre, XV = xylem vessels.

Soft wood stem sections of glasshouse-grown mature plants displayed similar characteristics to that of in vitro stems for both rootstocks. A prominent, continuous fascicular cambium about 4–6 cell layers in thickness could be observed in 'Reed' (Figure 1c), while it was narrow (2–3 cell layers) in 'Velvick' (Figure 1d), although more clearly present than for in vitro shoots. Sclerenchymatous phloem fibre was not observed at this proximity to the shoot tip in 'Reed' stem sections (Figure 1c). Again, a large amount of phloem fibre was present in 'Velvick', nearly forming a continuous ring at the inner boundary of cortex layers (Figure 1d).

### 3.1.2. Impact of Etiolation on Stem Morphology in Avocado

Etiolated 'Velvick' stem sections shared common features with in vitro 'Reed' stems, supportive a switch towards improved rooting competency. Mainly, the fascicular cambium was well formed with 4–6 layers of cells in etiolated shoots (Figure 2a). Consistent with this structure as a marker or requirement for rooting capacity, when the shoots were de-etiolated, this thick layer of fascicular cambium composed of rectangular cells could no longer be observed (Figure 2b).

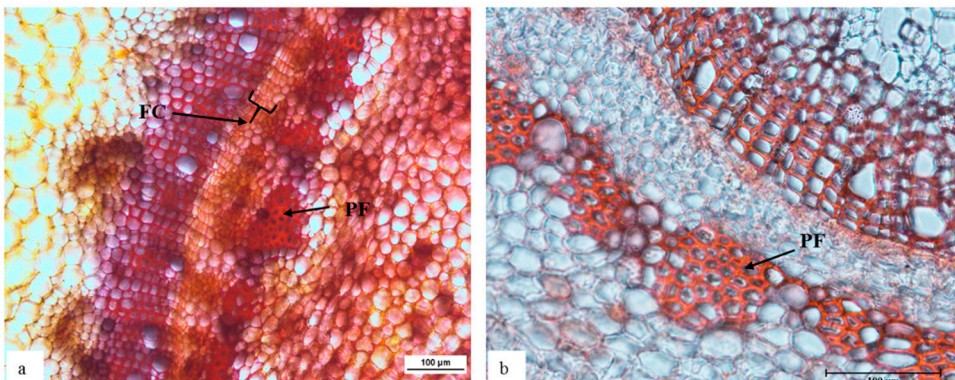

**Figure 2.** The shoot of rootstock 'Velvick' subjected to etiolation formed a thick and distinctive arrangement of cells as a continuous fascicular cambium (**a**), which was not identifiable after de-etiolation (**b**) of shoots. Transverse sections of 'Velvick' stem at 2.5 cm below the shoot tip: etiolated shoot (**a**), de-etiolated shoot (**b**). Stems were embedded in agar, sectioned at 50 μm, cleared with chloral hydrate and stained with Safranin O (*n* = 5). FC = fascicular cambium and PF = phloem fibre.

### 3.1.3. Morphology of Root Initiation in Avocado In Vitro Stems

The transverse sections of in vitro 'Reed' shoots treated for root induction were observed to identify the region of root initial formation in avocado (Figure 3). The root initials were present at the outer boundary of xylem tissue where thick fascicular cambium was present. Roots then extended outwards through the phloem tissues and cortex.

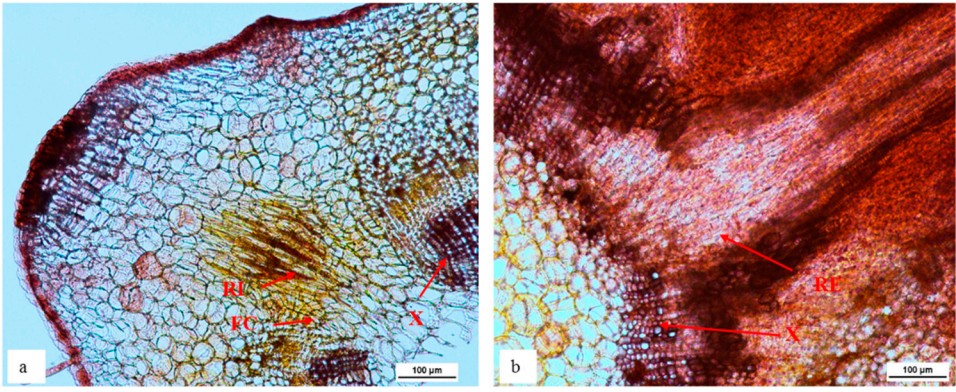

**Figure 3.** The root initial starting to form from the outer boundary of xylem and thick fascicular cambium layer (**a**) and extend outwards through the phloem (**b**) of stem in in vitro cultured 'Reed' during rooting. Transverse stem sections 4 weeks after rooting treatment from the base of stem tissue. Stems were embedded in agar, sectioned at 50 μm, cleared with chloral hydrate and stained with Safranin O (*n* = 5). FC = fascicular cambium, RI = root initial, RE = root extension, X = xylem.

### 3.2. Foliar Morphological Study of Avocado

Due to the role of leaves in controlling plant water potential, it was hypothesised that leaf micromorphology may contribute to differences in the acclimation success between avocado rootstocks 'Reed' and 'Velvick'.

#### 3.2.1. Stomates

In general plants under in vitro conditions possess lesser number of stomata and often remain fully open due to the very high (nearly 100%) relative humidity in the culture vessel [21,22]. Also, stomata developed under in vitro conditions can be non-functional even under stress [21,22]. Paracytic stomata were found on the abaxial surface of leaves in both avocado rootstocks 'Reed' and 'Velvick'. At the in vitro stage, the lower epidermis appeared to have completely open stomata in 'Reed' with maximum curvature of guard

cells (Figure 4a). However, at 5 days after de-flasking in water and 1 month under ac-climation, stomates appeared functional based on the pore size and curvature of guard cells (Figure 4b,c). In 'Velvick' stomates under in vitro condition appear to have more curvature in the guard cells compared to guard cells at 5 days after de-flasking in water and 1 month under acclimation (Figure 4d,e,f). The major epidermal difference between the two rootstocks was the size of epidermal cells at the in vitro, de-flasked and acclimated stages. At each stage, the epidermis of 'Velvick' was composed of a large number of smaller epidermal cells compared to 'Reed', which had fewer and large epidermal cells (Figure 4).

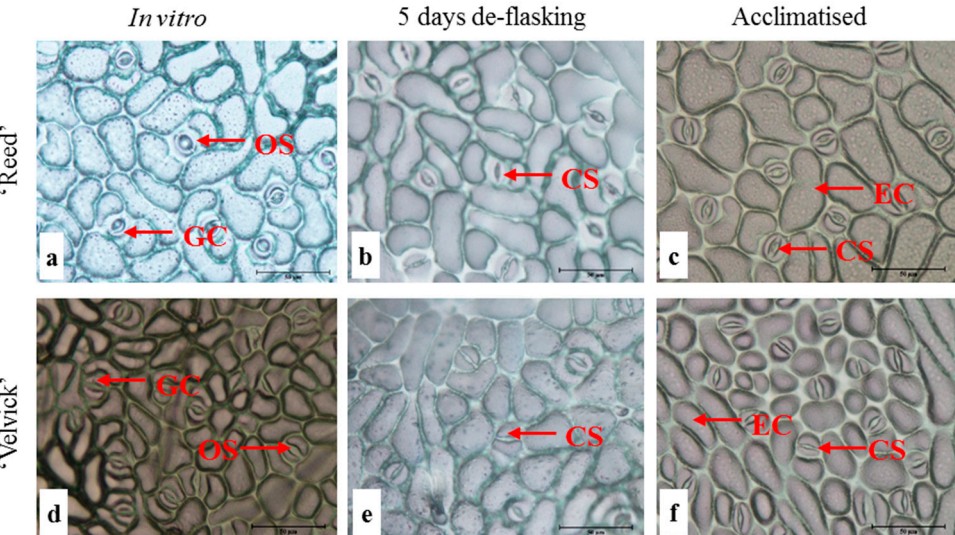

**Figure 4.** Stomates under in vitro conditions appear to be fully open compared to the de-flasked and acclimated stages in both rootstocks 'Reed' and 'Velvick'. Lower epidermis of the 3rd fully expanded leaf of in vitro cultured plantlets of 'Reed' (**a–c**) and 'Velvick' (**d–f**) at different stages, in vitro shoot before rooting (**a,d**), 5 days after de-flasking in water before acclimation (**b,e**) and 1 month in acclimation (**c,f**) were sampled (*n* = 5). OS = open stomates, CS = closed stomates GC = guard cells, EC = epidermal cells.

Stomatal index and stomatal density were compared between the rootstocks at in vitro rooted stage before acclimation and 1-month post acclimation. Also, these two parameters were compared between the two stages for individual rootstocks. A significantly higher stomatal index was recorded at in vitro stage for 'Reed' than 'Velvick' (*p* = 0.024) and no difference was found at acclimation stage (Figure 5A). Stomatal index significantly increased (*p* = 0.005) for both rootstocks during the transition from in vitro conditions to acclimation.

Stomatal density was significantly higher in 'Velvick' at both in vitro stage (*p* = 0.025) and 1 month into acclimation (*p* = 0.001) than that of 'Reed' (Figure 5B). However, upon acclimation, the stomatal density increased 6-fold in 'Velvick' (*p* < 0.001), while 'Reed' showed no significant increase in stomatal density.

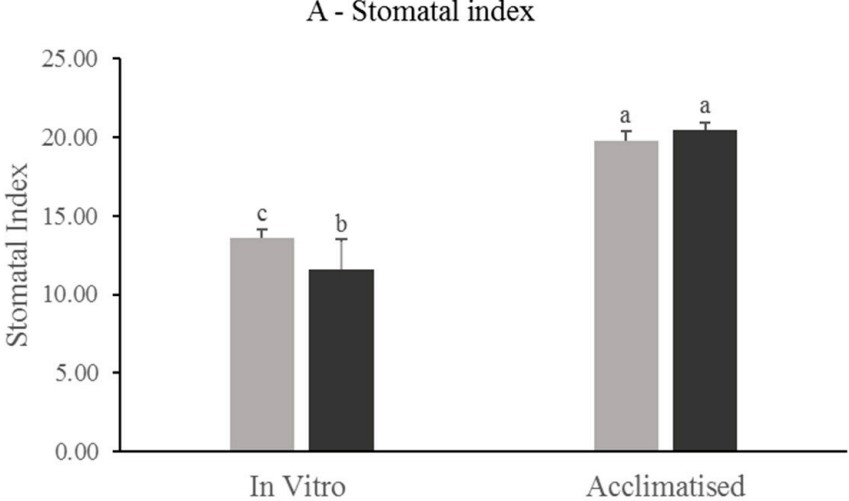

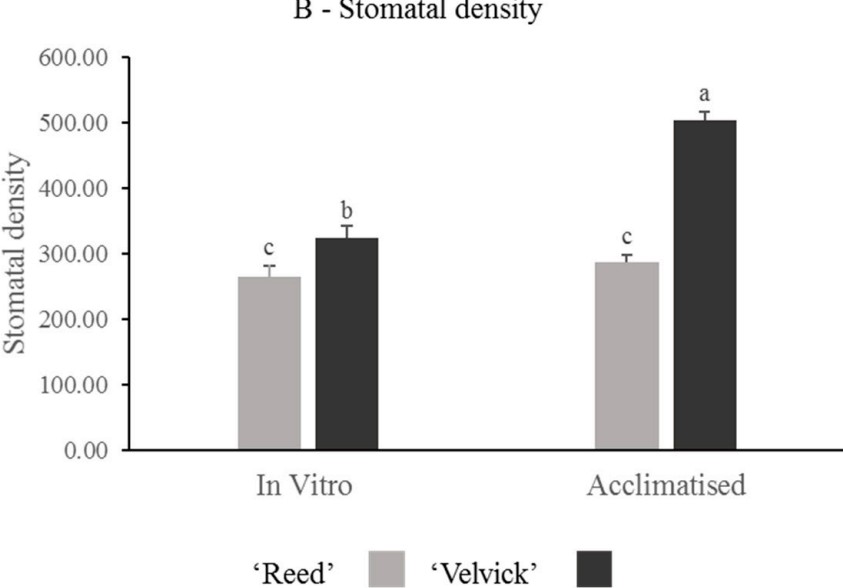

**Figure 5.** The difference in stomatal index (**A**) and stomatal density (**B**) of avocado rootstocks at in vitro and acclimated stages, as well as changes occur in individual rootstocks during transition. Stomatal index between 'Reed' and 'Velvick' was significantly different only at in vitro stage (**A**), but stomatal density was significantly different at both in vitro and 1 month into acclimation (**B**). For 'Reed', stomatal index was significantly increased during the transition from in vitro to acclimation, whereas for 'Velvick' both stomatal index and stomatal density are increased. The letters a, b and c indicate significantly different values within a tested stage between two rootstocks and between the two stages (in vitro and acclimated) at confidence level $p = 0.05$, $n = 10$ with two replicates and error bars indicate standard error.

### 3.2.2. Trichomes

Both unicellular and multicellular trichomes were found on both the upper and lower epidermis, but comparatively more on the abaxial surface in both rootstocks. Trichomes were predominantly located around veins, though they were distributed all over the leaf surface (Figure 6).

The amount of trichomes drastically differed between the two rootstocks at the in vitro stage as well as in the acclimation stage. Trichome density was significantly higher in 'Velvick' compared to 'Reed' at both stages ($p < 0.001$) (Figure 7). Nevertheless, in both

rootstocks, the progression from in vitro to acclimated stage increased the trichome density significantly ($p < 0.001$).

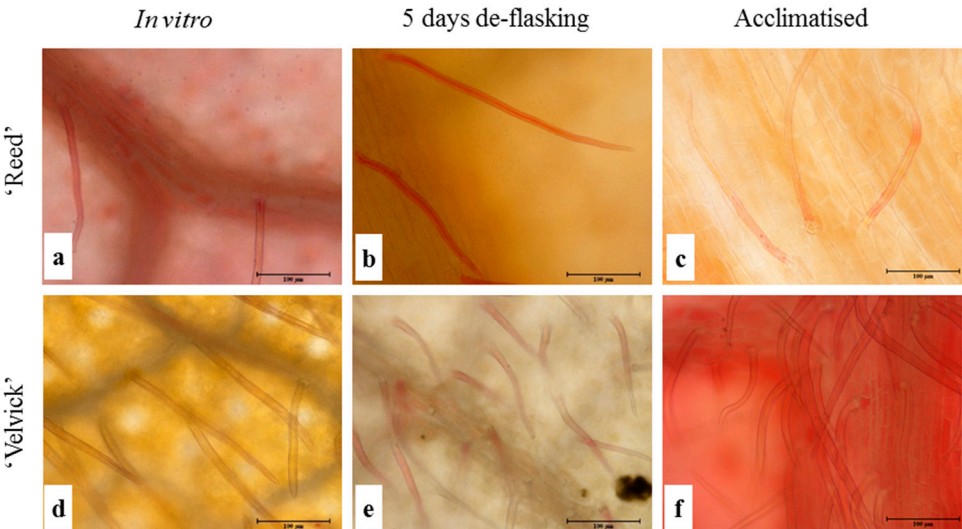

**Figure 6.** Trichomes were more abundant on lower epidermis of 'Velvick' than in 'Reed'. Lower epidermis of the 3rd fully expanded leaf of in vitro cultured plantlets of 'Reed' (**a**–**c**) and 'Velvick' (**d**–**f**) at different stages, in vitro shoot before rooting (**a**,**d**), 5 days after de-flasking in water before acclimation (**b**,**e**) and 1 month in acclimation (**c**,**f**) were sampled ($n = 5$) to observe trichomes.

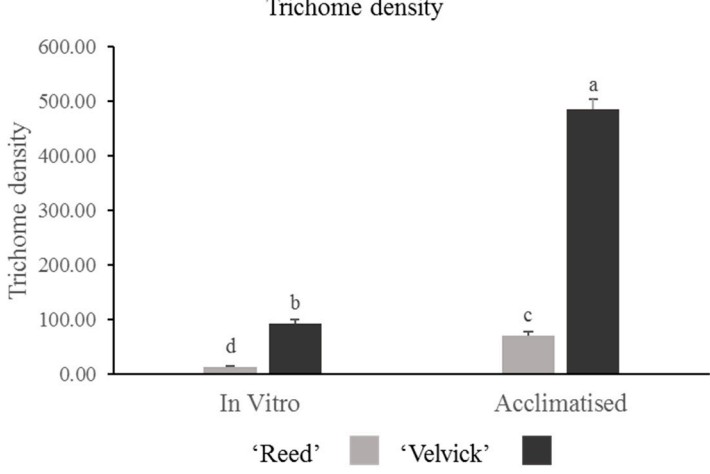

**Figure 7.** Trichome density was significantly greater in 'Velvick' than 'Reed' at both in vitro and acclimation. Both the rootstocks increased the trichome density from in vitro to acclimation stage. The letters a, b, c, and d indicate significantly different values within a tested stage between two rootstocks and between the two stages (in vitro and acclimated) at confidence level $p = 0.05$, $n = 10$ with two replicates and error bars indicate standard error.

### 3.2.3. Vein Architecture

Leaf architecture was observed to identify vein-islets and vein termination in fixed leaves of the two rootstocks at in vitro and acclimation stages (Figure 8).

Quantitative analysis of vein-islet density revealed that in vitro and acclimated leaves of 'Velvick' had significantly more vein islets than leaves of 'Reed' ($p < 0.001$) (Figure 9A). The transition from in vitro to acclimation did not change the vein-islet density of 'Reed', though a significant increase of vein islet density was recorded with 'Velvick' ($p < 0.001$) (Figure 9A). Under in vitro conditions, 'Reed' possessed significantly more vein terminations ($p < 0.001$), though this was overtaken by 'Velvick' after acclimation ($p < 0.001$) (Figure 9B). Consistent with this, the transition from in vitro to ex vitro stage reduced the amount of vein terminations in 'Reed' ($p < 0.001$) but increased in 'Velvick' ($p < 0.001$) (Figure 9B).

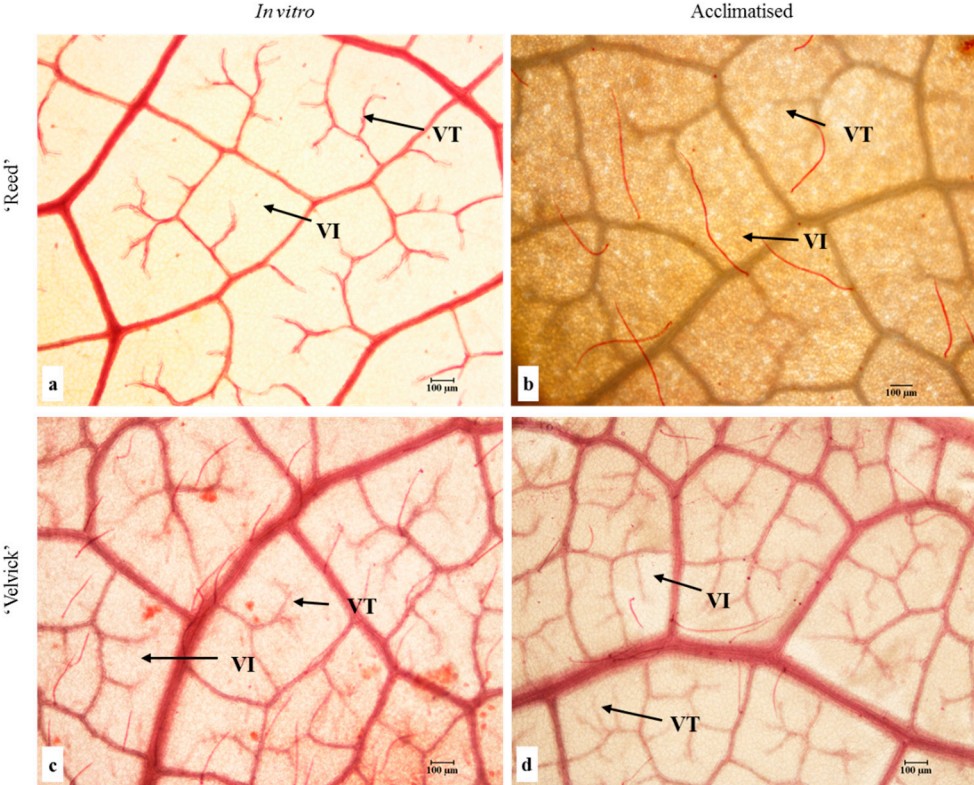

**Figure 8.** Leaf vein architecture of in vitro (**a**,**c**) and acclimated (**b**,**d**) tissue culture plantlets of 'Reed' (**a**,**b**) and 'Velvick' (**c**,**d**). Trichomes are also stained in red. VI = vein-islet (smallest area of photosynthetic tissue surrounded by conducting strands) and VT = vein termination were observed in tissues fixed in a 1:1:3 (*v:v*) solution of formalin, acetic acid and ethanol, bleached in NaOH and cleared with chloral hydrate.

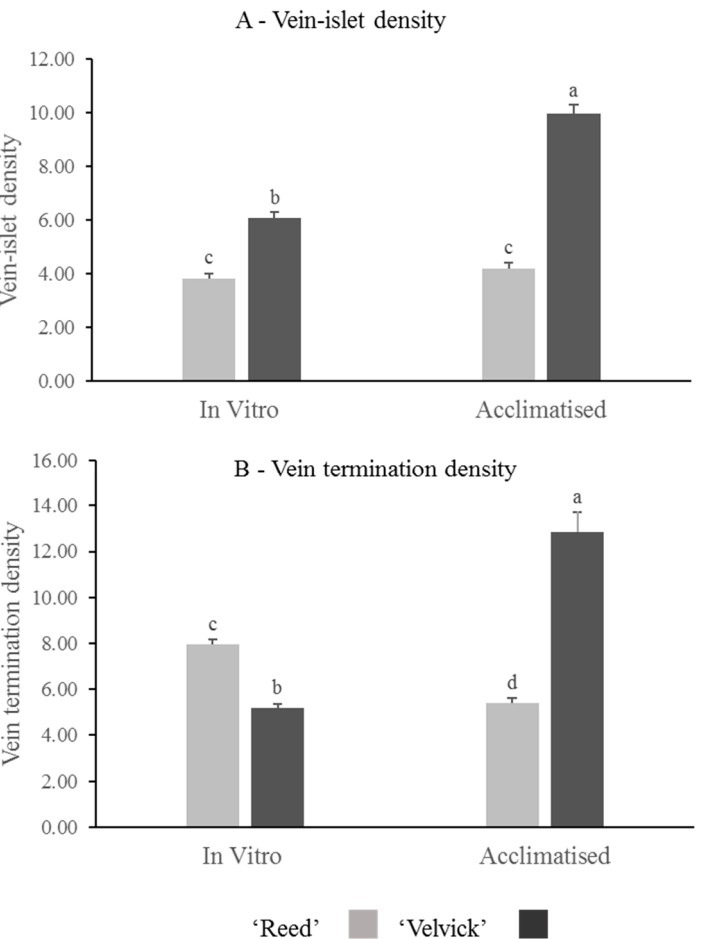

**Figure 9.** The difference in vein-islet density (**A**) and vein termination density (**B**) of avocado rootstocks at in vitro and acclimated stages, as well as changes occur in individual rootstocks during transition. 'Velvick' had significantly higher vein-islet density in both in vitro and acclimated leaves compared to 'Reed'. Vein termination density was higher in 'Reed' at in vitro stage but lower at acclimated stage compared to 'Velvick'. Vein islet density remained unchanged from in vitro acclimation with 'Reed', but vein termination density was significantly reduced. For 'Velvick' both vein-islet density and vein termination density were significantly increased. The letters a, b, c, and d indicate significantly different values within a tested stage between two rootstocks and between the two satges (in vitro and acclimated) at confidence level $p = 0.001$, $n = 10$ with two replicates and error bars indicate standard error.

### 3.3. Root Structure of In Vitro Cultured Avocado

Visual assessments reveal that in vitro regenerated roots of 'Reed' are thin, whereas 'Velvick' roots are thick (Figure 10).

Transverse sections taken 1 cm from the root tip of de-flasked in vitro plantlets for both rootstocks had distinguishable anatomical differences (Figure 11). Fully differentiated large xylem vessels were abundant in the center of 'Reed' roots (Figure 11a,b). In contrast, 'Velvick' had very few large xylem vessels (Figure 11c,d). Secondary cell wall thickening was observed in the endodermis of 'Reed' (Figure 11a), but not within the endodermis of 'Velvick' (Figure 11b). A wide root cortex composed of parenchymatic cells was present in 'Velvick', where it was narrow in 'Reed'. Phloem tissues could be observed in cross section of both rootstocks, but phloem fibre was not noticeable in roots.

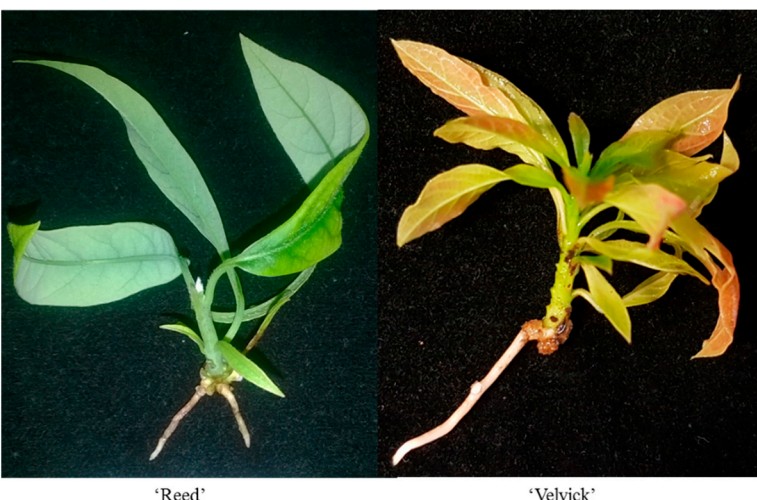

**Figure 10.** Differences in size of roots of the two different rootstocks, thin roots of easy-to-root 'Reed' and plump root of difficult-to-root 'Velvick'.

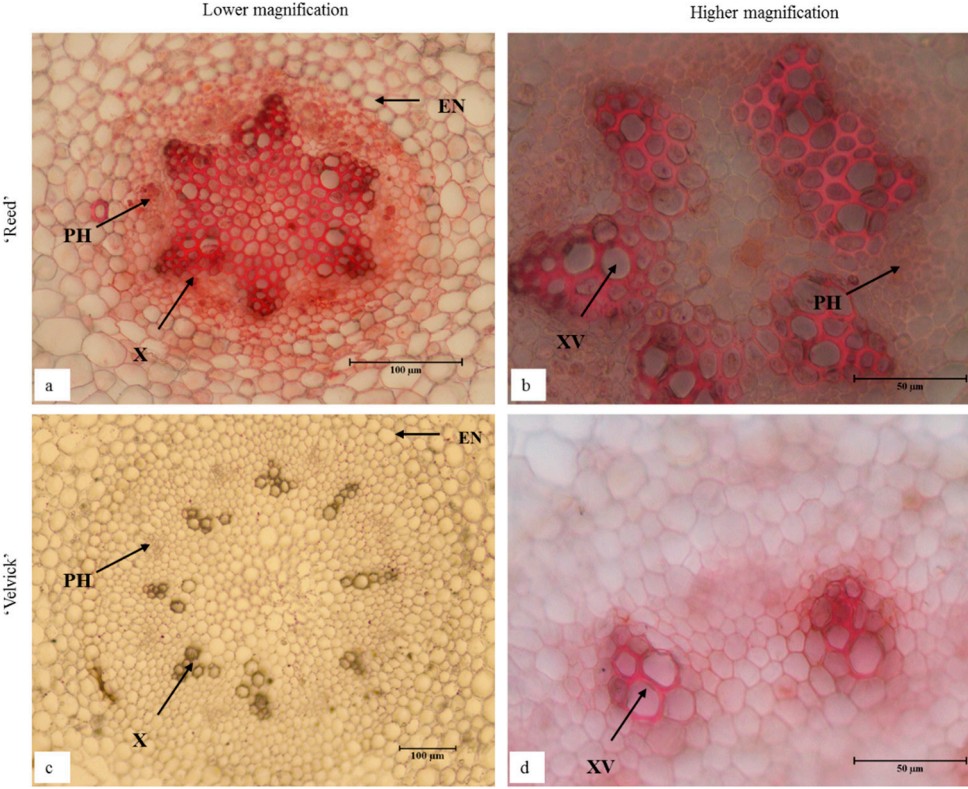

**Figure 11.** In vitro regenerated roots of 'Reed' (**a**,**b**) and 'Velvick' (**c**,**d**) possessed distinguishable anatomical difference in transverse sections. Fully differentiated large xylem vessels were abundant in the center of 'Reed' roots (**a**,**b**) but 'Velvick' had very few large xylem vessels (**c**,**d**). Secondary cell wall thickening was observed in the endodermis of 'Reed' (**a**), but not within the endodermis of 'Velvick' (**c**). Sections were cleared with chloral hydrate and stained with Safranin O (*n* = 3), X = xylem, XV = xylem vessels, PH = phloem and EN = endodermis.

## 4. Discussion

Adventitious roots are roots formed from non-root tissues, which in many species play important natural roles for survival under stress conditions such as waterlogged environments or on aerial tree canopies [23]. The formation of adventitious roots is an energetically expensive and complex process comprised of 3 distinct stages: (1) root initiation,

(2) emergence and (3) elongation [24–26]. Inhibition of any one of these stages can lead to loss of adventitious rooting competency. Woody plants often lose their rooting potential with age though it can be regained through specific treatments such as dark incubation or heavy pruning [9,12,18,27]. This is a major issue for many outcrossing horticultural and forestry crop industries, where adventitious root formation on shoot cuttings taken from mature trees is essential for clonal propagation. Avocado is once such species where rooting is the major bottleneck to propagation of elite rootstock varieties, calling for the need for improved propagation technologies. Tissue culture is one such technology being developed for avocado propagation, which offers huge advantages in throughput and efficiency of clonal plant production, but where again adventitious root induction has proven a major bottleneck for tissue cultures initiated from physiologically mature material [4].

The two avocado rootstocks used in this study belong to different ecological races; 'Reed', a Guatemalan [28,29] and 'Velvick', a West Indian [30,31]. Our previous work in rooting these two rootstocks in tissue culture revealed that, though the rooting treatment optimization is difficult, 'Reed' is a relatively easy-to-root rootstock with 100% rooting success, while 'Velvick' is very difficult-to-root recording only 33% rooting success. This variation in rooting competency among rootstocks is not confined only to avocado, but observed in other woody perennials such as apple (*Malus domestica*) [32], olive (*Oleas europaea sativa* L.) [33] and pear (*pyrus communis* L.) [34]. Anatomical differences that may contribute to such rooting variations have been reported for *Quercus bicolor*, *Quercus macrocarpa* [35], *Hedera helix*, *Grevillea paniculacea* [36] and *Corymbia torelliana x Corymbia citriodora* [37].

In this study, it was hypothesised that the micromorphology of in vitro shoots of 'Reed' and 'Velvick' may provide insights into their rooting variabilities in culture. Supporting this hypothesis, the two avocado rootstocks possessed distinguishable differences with respect to the cellular arrangement and type of tissues present in their stem structure, where adventitious roots are induced. A broad and continuous fascicular cambium, xylem rays directly opening to the cortex and no or very few phloem fibres were observed in both in vitro and glasshouse grown soft wood stems of easy-to-root rootstock 'Reed'. These may be structural features of this rootstock that facilitate ease of adventitious root induction. In contrast, the difficult-to-root 'Velvick' lacked the thick layer of continuous fascicular cambium but had highly abundant phloem fibres nearly forming a complete ring at the inner cortex of both in vitro and glasshouse samples. Therefore, we propose that the above structural differences may be the reason for the difference in rooting abilities of the two rootstocks studied. Consistent with this, Maynard and Bassuk [16] identified large amounts and a continuous ring of lignified phloem fibres and less cambium as features of difficult-to-root *Carpinus betulus* L. fastigiata. Also, vascular lignification and sclerification have reported to be directly related with poor rooting in *Corymbia torelliana × Corymbia citriodora* [37]. The presence of nearly a complete ring of phloem fibre is thought to act as a physical barrier for root extension [16,38], hence disadvantages for adventitious rooting. Meanwhile, many layers of cambium cells and minor secondary thickening in phloem and xylem rays directly opening to cortex were characteristic features of easy-to-root plants [16,17,38].

Work done by John [39] with hybrid larch (Larix × eurolepis Henry) and Harbage et al., [40] with apple, revealed that root primordia arise in tissue produced either side of the fascicular cambium at the outer xylem boundary or inner phloem boundary. Our observation of stems treated for root induction in 'Reed', also revealed root initiation at the boundary of the fascicular cambium and xylem in avocado. Another study by Junyan et al. [38] with apple micro-cuttings explains that meristemoids, which are localised group of cambium like cells that has differentiation ability, and are formed as a result of fascicular cambium activity upon root induction treatments, emphasising the necessity of prominent cambium tissue for this purpose.

The above arguments for morphological requirements, or markers, of rooting competency, were also strengthened when we looked at the etiolated and de-etiolated stem sections of hard-to-root rootstock 'Velvick'. Etiolation is a process that causes many morphological and molecular changes within plant tissues [41]. It is thought to promote, or

rejuvenate, the capacity for rooting in recalcitrant species [42] by increasing sugar content, reducing auxin inhibitory flavonoids [43] and structural alterations [17]. Etiolation appeared to promote the formation of a multi-layered fascicular cambium with a discontinued lignified phloem fibrous barrier in 'Velvick' stem cuttings. Moreover, these morphological features appeared to be lost upon de-etiolation, or exposure to light conditions for 8 weeks. We therefore suggest that the attainment of these specific features in etiolated stems that are then lost upon de-etiolation explains the attainment and loss of rooting function in response to etiolation.

The acclimation process is generally challenging for in vitro cultured plantlets due to the lack of cellular support structures to maintain water content within the plantlet [44]. Under in vitro conditions, plants are semi autotrophic and lack structures such as a fully developed cuticle, functionally differentiated photosynthetic tissues and roots, and functional stomates [45–47]. We previously observed that tissue cultured 'Reed' plantlets were readily acclimated using a simple protocol with no specialised conditions [15]. However, this acclimation procedure resulted in very poor survival rates with 'Velvick', which has proven to be very difficult-to-acclimate. We thus sought to determine any correlation between plantlet morphology and acclimation success in these rootstocks.

During our research in avocado micropropagation it was found that neither the number of roots nor root length was correlated to acclimation success of 'Velvick' and 'Reed' (data not shown), which recorded 30% and 97% acclimation success with the optimum rootstock specific acclimation practices. Meanwhile, the leaf epidermal structure appeared to influence acclimation. Under microscopic inspection it was observed that 'Velvick' had small epidermal cells, thus could have resulted in high surface area to volume ratio. Contrastingly, the 'Reed' epidermis was composed of relatively large epidermal cells with a possibility to have a low surface area to volume ratio. One theory is that this may contribute to increased water loss from the 'Velvick' leaf surface, resulting in poorer acclimation capacity relative to 'Reed'. Furthermore, the stomatal index was significantly lesser in 'Velvick' compared to 'Reed' but stomatal density was higher in 'Velvick' compared to 'Reed' tissue cultured plantlets. The higher number of stomata could aggravate water loss, particularly if they are not fully functional at the start of acclimation, which can be a possible reason for poor acclimation success observed in 'Velvick'.

Trichome density and distribution are suggested to be important for maintaining a high relative water content in both in vitro and ex vitro plants [48]. The relatively high number of trichomes in 'Velvick' suggest that it has an inherent structural barrier to water loss from leaf surfaces. This is counter-intuitive to the hard-to-acclimate nature of 'Velvick' and suggests that other factors may be more relevant to controlling acclimation stress in 'Velvick' in vitro plantlets.

The area of photosynthetic tissue encompassed by vein strands is referred to as a vein-islet. The two fundamental functions of fully formed veins are to translocate water and carbohydrates and provide mechanical strength to leaves [49]. Therefore, it can be hypothesised that the density of vein-islets and vein terminations to be a measure of plants' efficiency to maintain high water potential within the leaf during acclimation. Our results showed that in vitro leaves of 'Velvick' had a significantly higher vein-islet density but significantly less vein terminations compared to 'Reed'. Increased vein-islet density from in vitro to ex vitro transition and branched vein terminations are reported to be a common feature for in vitro cultured plantlets [50]. In the case of these avocado rootstocks, increased vein-islet density was not associated with improved acclimation rates. However, we hypothesise that the increased number of vein terminations on the leaf surface of 'Reed' may contribute to its improved acclimation success by maintaining efficient water supply to leaves.

The water conductance of a plant is maintained not only by stomatal conductance but also by efficient water absorption and transport from the roots [51]. The root transverse sections obtained for the two rootstocks revealed profound differences in their vascular system. Large and fully differentiated xylem tissue was the distinctive feature of 'Reed'

roots in comparison to 'Velvick'. In our opinion, the fully differentiated xylem structure of 'Reed' may enable more efficient transport of water to maintain high water potential in the shoot and compensate for the lack of cuticle and poorly functioning stomates in in vitro propagated shoots. Meanwhile, the lack of well-developed abundant xylem vessels in 'Velvick' may not adequately maintain water potential within the shoot during applied acclimation conditions. Consistent with this, acclimation failure of additional species in vitro has been related to a reduced ability to maintain water and ion relationships as a result of partially or under developed secondary xylem in roots. The study by Soukup et al. [52], compared the anatomical changes of in vitro cultured wild cherry (*prunus avium* L.) and oak (*Quercus robur* L.), illustrating benefits of early lignification of root xylem in oak for better plantlet survival during acclimation. Another study on leaf water potential in tissue cultured grapevine plants revealed that it is better to have zero roots than to have poorly developed roots for maintaining water potential in tissue cultured plantlets [51]. In that study, shoots immersed in water after removal of the in vitro regenerated root system maintained better water potential and had no symptoms of wilting compared to shoots with an intact root system. This further indicates the negative influence of inefficient root system micromorphology on the acclimation process.

This study for the first time presents a histological basis for inter-rootstock variation in rooting and acclimation success of in vitro cultured avocado. Our investigations, comparing leaf, stem and root structures of 'Reed' and 'Velvick', posits the possible morphological requirements for adventitious rooting and acclimation success in tissue cultured avocado. Identification of structural features directly linked to rooting and acclimation will undoubtedly facilitate further improvements to the avocado micropropagation. Variation in rooting and acclimation is a frequent problem in other economically important woody crop species. The identification of anatomical barriers to rooting and acclimation enable to use of corrective strategies; e.g., use of plant hormones to induce fascicular cambium, longer rooting periods to enable fully developed xylem. Therefore, we believe that the knowledge generated in this study will provide clues to solve similar research problems with other woody plants.

## 5. Conclusions

The anatomy of the stem of in vitro regenerated avocado shoots directly impacts the rooting ability. Presence of a well formed, continuous fascicular cambium and less abundant phloem fibres are indicators of high rooting potential. Difficulty in acclimation is suggested to be mainly linked with root anatomy, but also affected by leaf structure. The reduced amenability of 'Velvick' to acclimation conditions may be a combined effect of having fewer, well-differentiated root xylem vessels, smaller epidermal cells, high stomatal density and a reduced vein termination density, culminating in high water stress levels upon removal from in vitro conditions.

**Author Contributions:** J.H.-B.: conceptualization, methodology, validation, formal analysis, investigation, data curation, writing—original draft preparation, visualization; A.H.: review and editing; N.M.: resources, review and editing, project administration, funding acquisition. All authors have read and agreed to the published version of the manuscript.

**Funding:** This research was funded by the Australian Research Council Linkage Programme (grant no. LP130100870) (2013), Advance Queensland Innovation Partnership Programme (grant no. 9739018-01-155-21-021268) (2017), the Department of Agriculture and Fisheries, Australia and the University of Queensland, Australia collaborating with industry partners; Primary Growth Pty Ltd., Jasper Farms Holdings Pty Ltd., Millwood Holdings Pty Ltd., T/A Delroy Orchards and Anderson Horticulture Pty Ltd., J. Hiti-Bandaralage was supported by an Australian Postgraduate Award.

**Informed Consent Statement:** Not applicable.

**Data Availability Statement:** Not applicable.

**Acknowledgments:** A special thanks to Anderson Horticulture Pty Ltd. for assisting the preparation of etiolated 'Velvick' material for experiments.

**Conflicts of Interest:** The authors declare no conflict of interest. The funders had no role in the design of the study; in the collection, analyses, or interpretation of data; in the writing of the manuscript; or in the decision to publish the results.

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
