# Peer review of "Structural Disparity of Avocado Rootstocks In Vitro for Rooting and Acclimation Success"

_2037-0164, doi:10.3390/ijpb13040035_

Round 1
Reviewer 1 Report
General: Some improvements to language still possible. When possible, please use plantlet/s when referring to TC material, in place of plant/s. The names used in refencing in text don't coincide with the reference number. Authors seem to have not used reference software, so a thorough check is needed. For example you use a reference on teak and talk about grapevine. My main concern is why authors used only one rootstock - Velvik for etiolation and presented in Fig. 2? A comparison with Reed would have been useful. Similarly, in Fig 3 only Reed is shown but no comparison with Velvick.
I have given suggestions in attached PDF. Please check.
TITLE: Please consider revising as you have compared the anatomy of in vitro material with greenhouse plants.
The abstract requires minor revision. I have suggestions in the attached PDF. Introduction:
Please revise sentence lines 54-56. Suggest "At the same time, stress factors under in vitro conditions can alter the inherent morphological features in plants, including essential structural characters vital for survival during rooting and acclimation". Please try to be clear what you want to say.
Line 65 - Please rephrase
M&M:
As all the readers are not familiar with avocado rootstock propagation, some explanation of why 'etiolation' is important at some point (either in Introduction or in M&M) would help.
2.1.1 - observations
112 - deflasking stage - how many subcultures, period between subcultures etc. What were the growth conditions of cultures?
113 - 115; which side of the leaves are we counting here?
124 - 10 samples = 10 plants?
103 - continue sentence
Line 82 - give the references after "Hiti-Bandaralage et al"
84 - remove Mature.
81-87: While it maybe immaterial for in vitro cultures, the time of sampling for glasshouse material matters. Please give the month and some environmental (e.g. day length) parameters.
137 - why v:v for 70% ethanol. You haven't even mentioned about water. If you want to use w:v for NaOH , suggest saying it is an aqueous solution.
Results
176-77 - Why a reference has been inserted in this section?
209 - 215 - This is part of your research hypothesis. It should be in the introduction, and then in Discussion.
Fig. 2. Why a TS of Reed not presented? Will ad value to the paper if a comparison is made.
fig. 3. Why rooting in vitro shoot of Veverik not presented? This will add to the value of the paper.
fig. 5a - Columns on left - b and c should switch places.
288 & 311 'duplicated samples' - what are these? Don't understand. Is this relevant?
321 - chloral with 'c' lower case.
Fig. 9B - consider revising the positions of a, b, c and d on the columns.
339 -341 - Are you comparing individual root stock or both? If both, as apparent from rest of the sentence, why use 'for an individual rootstock' at the end of sentence? This is confusing.
The authors should consider combining some figures.
248 - easy
354 - consistency in naming cultivars needed. At times cv 'Reed' . In other places without cv! Please check throughout.
360 - possessed had - use only one
365-66 - no need to mention scale bar sizes as it is given in the picture.
Discussion
382- what is meant by mature cultures? Please elaborate.
401, 404, 419, 426, 431, 433 and further down, marked. No need to refer to figures here. We have already seen this in results.
415, 416 wrong references.
420 - 35 is not Junyang!!!
PLEASE check all references as what you refer are not the relevant one. e.g. 491 - Soukap is not ref # 49!
Reference 48 is not grape but teak!
447 - values respectively for the two rootstocks?
507 - 508 - You argue that the knowledge gained will help to solve the problem. Once you know that the morpho-anatomical features are disadvantageous, what can one do to solve this problem? So, in your study, you explain the reasons for poor rooting and acclimation. Science is knowledge, so your work is valuable. I can't however comprehend how your research can solve the problem Please rephrase.
References
Please give a thorough check and revise.

Reviewer 2 Report
The paper entitled: “Structural disparity of avocado rootstocks in vitro for rooting and acclimation success”, discusses concentrated on (1) the histological analysis to offer a potential explanation for variations in the capacity of two avocado rootstocks (‘Reed’, and ‘Velvick’) to root and acclimate when grown in tissue culture, (2) Comparing the structural characteristics of each cultivar's acclimatizing plants and in vitro shoots.
The paper falls perfectly within the scope of the International Journal of Plant Biology (IJPB), However, some minor corrections are needed before considering your paper for publication.
- The abstract is well written linguistically (despite some words that needs to be replaced). However, I think the abstract of this study needs to be made clearer, the objectives of this study as well.
- In the introduction, please add more information about Avocado’s propagation, mainly, about the etiolation of the species.
- In the introduction, add the scientific name and other important information (that are missing) of the studied plant (Avocado, Persea americana Mill., family Lauraceae …)
- Line 54-56: try to rephrase these sentences.
- In Materials and Methods, authors must include some of the climatic conditions in the glasshouses (temperature, humidity, day/night length…).
- Line 142: change to “Sections of 2 - 3 mm2”
- Consider writing the Latin words in Italics (e.g. in vitro, ex vitro, in vivo …)
- Line 348: “easy-to-root”
- Lines 495-496: clarify this sentence.
- No need to insert figures’ callouts in the discussion. Consider removing them.
- “Variation in rooting and acclimation is a frequent problem in other economically important woody crop species, therefore, we believe that the knowledge generated in this study will provide clues to solve similar research problems with other woody plants.”, How can that be possible? The current research is important as it explains a lot about the poor rooting observed in many Avocado cultivars, but how can we face that problem? Can the authors explain more their assertions.
The authors must insert the references automatically respecting IJPB referencing style.
